# Heavy Metals Enrichment Associated with Water-Level Fluctuations in the Riparian Soils of the Xiaowan Reservoir, Lancang River

**DOI:** 10.3390/ijerph191912902

**Published:** 2022-10-08

**Authors:** Ronghua Zhong, Yun Zhang, Xingwu Duan, Fei Wang, Raheel Anjum

**Affiliations:** 1Institute of International Rivers and Eco-Security, Yunnan University, Kunming 650091, China; 2Institute of Mountain Hazards and Environment, Chinese Academy of Sciences, Chengdu 610041, China; 3Department of Economics, Abdul Wali Khan University, Mardan 23200, Pakistan

**Keywords:** heavy metals pollution, riparian zone soils, water-level fluctuations, Xiaowan Reservoir, Lancang River

## Abstract

The cascade hydropower development in the Lancang River has significantly modified the hydrologic regime and is consequently responsible for many local environmental changes. The influence of the altered hydrological regime on heavy metals accumulation in the soils of the riparian zone was evaluated for the Xiaowan Reservoir (XWR). Specifically, this study focused on investigating the trace metals As, Cd, Cr, Cu, Hg, Ni, Pb, and Zn and their concentrations in the riparian soils. Furthermore, this research aimed to examine the contamination levels of heavy metals by employing the geoaccumulation index (*I*_geo_) and the ecological risk index (*RI*), respectively. Additionally, the relationship between heavy metals and water level fluctuations as caused by the dam operation was explored. The results showed that heavy metals deposits occurred in relatively low levels in the riparian soils of XWR, even though several of these metals were revealed to occur in higher concentrations than the local background value. The *I*_geo_ assessment indicated that the riparian soils exhibited slight pollution by Hg at the Zhujie wharf (ZJW) and Cr at the transect of the Heihui River (HHR), and moderate contamination by As at ZJW. Moreover, the RI revealed that As in riparian soils is moderately hazardous while Hg poses a high risk at ZJW. The polluted water and sediments from upstream and upland of the riparian zone may be contributing to the changed concentrations of heavy metal in the riparian soils. The present study inferred that the WLFs due to reservoir impoundment play a vital role in the accumulation of trace metals in the riparian zone. However, more exhaustive investigations are necessary for verification.

## 1. Introduction

The presence of heavy metals is extensive within the water body, soils, and sediments in the river system [1,2,3,4]. Due to the rapid industrialization and urbanization along the river, trace metals in the environment are increasing [5,6]. Runoff and sediments containing trace metals relating to anthropogenic factors such as industrial events, agricultural activities, domestic garbage, and vehicle exhaust emission are continuously deposited in the valley’s environmental systems [7,8,9,10]. Metal contamination and migration of contaminants in the soil, together with hydrologic processes, are possible threats to the environment. These trace metals are continually accumulating in the riparian soils, sediments, and plants through biogeochemical processes, eventually exceeding the regional background concentration and leading to heavy metals pollution of the riparian zone [11,12]. Therefore, the riparian zone in the river system is subjected to growing pressure as a result of the hydroelectric development and reservoir impoundment [13]. Because of the connectivity of the river system, any changes to the upstream environment will inevitably have an impact on the ecological system and environment located downstream [14]. Thus, heavy metal contamination in the riparian zone, even an exceedingly low level, will cause severe damage to the regional eco-environment and river health [15,16,17]. Consequently, concentrations of heavy metals in soils and sediments are easily adversely affected by human activities [18,19] and water regulation [1,11,20,21]. In recent years, the pace of hydropower development has been accelerated on a global scale to meet the growing demand in energy consumption [22]. Furthermore, the progression of hydropower would aid in reducing air pollution and alleviating the effects of global climate change [22]. The operation of large dams and the impoundment of reservoirs have generated significant water level fluctuations (WLFs) in the riparian zone, which in turn could activate a series of eco-environmental responses [20,23,24,25,26]. Therefore, it is crucial to determine the influence of WLFs on heavy metal concentrations and its spatial distributions. Moreover, the contamination levels in the riparian zone need to be evaluated for ecological restoration and soil protection [27,28,29].

WLFs is an essential hydrological process for all rivers, lakes, and reservoirs [24,30]. The amplitude, frequency, and duration of WLFs have essential impacts on the physical processes (e.g., bank erosion, sedimentation patterns, and transparency) [31]. The changes in the water stage adjust the bank morphometry and influence the actions of erosion and sedimentation, essentially allowing WLFs to regulate the landform features and biogeochemical processes [32]. In consequence, WLFs dominantly regulate the structure and function of ecosystems along the riparian zone, which are formed by WLFs [23,25]. Considering that the riparian zone is shaped by WLFs, it is not surprising that these fluctuations are primarily responsible for regulating the structure and function of ecosystems throughout these riverine areas. Theoretically, the riparian zone refers to all areas of transition between aquatic and terrestrial environments, including the banks of rivers and lakes, flood zones, lateral benches, point bars, and islands, which are formed by the regular water level changes in unmanaged river channels or artificial water level control in lakes and reservoirs [24,33,34,35]. However, this study focuses on the riparian zone as it exists within reservoirs primarily produced by water level regulation, or otherwise known as the water level fluctuations zone (WLFZ).

As stated, there are two major types of WLFs, namely natural water level changes and artificial water level control [20]. The first type of WLFs may occur on changed spatial-temporal scales within natural flooding processes for all watersheds [24]. In contrast, the second type of WLFs is generated by the anthropogenic modification of the water levels in regulated lakes and reservoirs [11,20,31]. Furthermore, WLFs can be elicited by changes in the surface-wave field encouraged by wind force and ship traffic and can last from mere seconds to several hours [24]. The influence of WLFs is usually enhanced and exacerbated by climatic changes, including seasonal fluctuations in rainfall, atmospheric temperature, and evapotranspiration and significant modifications of the hydrological regime in the river basin system [24,25,31,36]. Nonetheless, water level regulations caused by climatic changes are typically identified as the natural hydrological regime and is characterized by relatively small rangeability and no seasonal variations [36]. In reality, hydrological processes have been artificially manipulated in regulated lakes and reservoirs, and the impact of WLFs is likely to be enhanced within the water level regulation when combined with accelerated global climate change [25]. Hence, WLFs would ultimately affect the entire riparian zone system, especially in large reservoirs [37]. 

Since the 20th century, increasing human activities have changed the dimensions and spatial-temporal distribution of floods to a great extent and have attracted increased attention regarding the consequences of WLFs prompted by anthropogenic disturbances [25,38,39]. The available information concerning the behavior of WLFs caused by human activity, is mainly related to the construction of hydraulic and hydropower plants for power generation, flood control, agricultural irrigation, and navigation improvement [11,20,40]. River damming and the impoundment of reservoirs have transformed the natural river flow regime into an artificial lacustrine system [20,41]. Many reservoirs have adopted the practice of impounding water during the dry season to the maximum level necessary for power generation while releasing the stocked piled water to the minimum level during the rainy season to facilitate flood control. The procedure is also known as “impounding the clear water and releasing the muddy water” and is responsible for the formation of severe WLFs [11,41,42,43]. WLFs can reach heights of tens of meters particularly in large reservoirs such as the Three Gorges Reservoir (TGR) and the Xiaowan Reservoir (XWR) in the upper Mekong River. These excessive WLFs could lead to an imbalance in both the aquatic and terrestrial ecosystems of the riparian zone, concerning the geomorphology and soil systems [20,44,45]. Unconventional WLFs induced by damming and reservoir impoundment, have a significant influence on the vegetation distribution, species diversity, soil nutrients cycle, and soil trace metals concentrations [1,20,27,32,44,45]. Additional biogeochemical processes in the riparian zone showing the effects of extreme WLFs include bank erosion and sedimentation [24,25,32].

XWR is the “dragon head” reservoir of the cascade hydropower projects located in the center and lower areas of the Lancang River (LCR). Stemming from a full impoundment of XWR in 2012 is a non-natural riparian zone with a variation of 60 m in vertical height that originated due to the regulation of the water level. Issues relating to the considerable ecological and environmental changes as a result of extreme WLFs have garnered augmented attention from the reservoir management department, local government and scientists. In recent years, several studies have reported on the geographical distribution of heavy metal concentrations and pollution levels in the riparian sediments of XWR [17,46,47,48]. Further comprehensive scientific research is necessary regarding the degradation of the eco-environment in the riparian zone of the XWR with a particular focus on the effect of WLFs on the soils. Therefore, it is prudent to determine the trace metals concentrations present in the riparian soils, and the level of contamination in relation to WLFs in XWR. In this study, the geoaccumulation index (*I*_geo_) combined with the potential ecological risk index (*RI*) were employed to evaluate the contamination status of heavy metals in the riparian soils of XWR. Specifically, the objectives were to (1) investigate the redistribution and accumulation of heavy metals in the riparian soils of XWR; (2) evaluate the pollution levels of heavy metals in soils using the geochemical approaches of *I*_geo_ and *RI*; and (3) deduce the relations between the detailed WLFs processes and heavy metals accumulation in soils of the riparian zone.

## 2. Materials and Methods

### 2.1. Study Area

The Xiaowan dam is located at the junction between the Fengqing and Nanjian counties of Yunnan Province, about 1.5 km downstream of the intersection between the Lancang River and Heihui River (Figure 1). The Xiaowan hydropower station is the second largest power station in the upper Mekong basin, with a crest length of 892.8 m and a dam height of 294.5 m, with an installed capacity of 4200 MW. The Xiaowan Dam construction started in January 2001 and its impoundment was completed from December 2008 to 31 August 2012. The backwater length of XWR is 179.6 km near the Gongguoqiao Dam site in the Lancang River and 125.3 km near the Xucun Dam in the Heihui River, respectively. The XWR covers an area of 189.1 km^2^ over eight districts and is mainly distributed through Fengqing, Changning, Longyang, and Nanjian counties. The total storage capacity of XWR is about 150.00 × 10^8^ m^3^ with a standard water level of 1240 m above sea level and its regulated storage capacity is 99.00 × 10^8^ m^3^. The XWR region is situated in the southwest mountainous gorge region of China, where the elevations of many mountains close to the XWR are higher than 2300 m, and the mainstream valley gradient ratio is over 15‰. The regional climate is a semi-tropical monsoon, with the annual average temperature ranging between 14.3 and 19 °C and annual precipitation ranging between 770 mm and 1330 mm, in different areas [49].

### 2.2. The Hydrologic Regime of XWR

Although power generation is the primary aim of the Xiaowan project, extensive added benefits presented itself, namely flood control, sediment trapping, and navigation. According to the annual operation program of XWR, a typical hydrological year consists of five stages: (1) P1, the water level drops gradually from 1240 m above sea level to 1180~1186 m above sea level from January to March due to reservoir emptying and drought; (2) P2, the water level is maintained at an approximately constant minimum level of 1180 m above sea level from April to May, to discharge the muddy water; (3) P3, the reservoir controls the water to keep the level at lower than 1232 m above sea level during June to mid-September, and then increase the water level to 1232~1240 m above sea level between mid-September and late October (P4); and (4) P4, the reservoir is fully impounded with a constant maximum level of approximately 1240 m above sea level during November to December (Figure 2). As a result, an artificial hydro-fluctuation belt (i.e., WLFZ) with a maximum vertical elevation difference of 60 m has been created. Owing to the changes in the river’s longitudinal elevation, the vertical height of WLFZ gradually decreases from 60 m at the dam site to 0 m at the terminal of the backwater that forms a natural river channel. Heretofore, the riparian zone of XWR has experienced seven full alternations of drying and wetting.

### 2.3. Soil Sampling and Chemical Analysis

Field sampling was conducted in May 2017 when the entire riparian zone was fully exposed. Six representative transects, specifically, near the Xiaowan Dam (XWD) site, Mangjie wharf (MJW), Manshui bridge (MSB), Yongbao bridge (YBB), Heihui River (HHR) and Zhujie wharf (ZJW), along XWR were selected as the sampling sites (Appendix A). At each cross-section, sequential sampling plots with 1 × 1 m grids were designed according to the topographical changes. Specifically, one sampling plot was selected for every elevation range with a vertical height of 5 m. Sampling plots included the complete lateral riparian zone from the maximum to minimum water level. A plastic shovel was used to sample soils in the riparian zone (0 to 10 cm). In every plot, soils from five points were sampled randomly to obtain a composite sample. In total, 87 composite soil samples were collected, placed in polyethylene bags, and sealed. The samples were then transferred to the lab, and immediately air-dried at room temperature. The dried soil was disaggregated, sieved using a 2 mm sieve, and subsequently resealed in polyethylene bags.

Before trace metal concentrations could be determined, digestion was performed with concentrated HCL-HNO3-HF-HClO (i.e., 5 mL HCL, 15 mL HNO3, 10 mL HF, and 5 mL HClO; HF represents the hydrofluoric acid) for all tested soils [50]. All the acids used in the digestion process were guaranteed reagent level (GR, 99.8%). After soil digestion, the concentrations of As, Cd, Cr, Cu, Ni, Pb, and Zn were measured utilizing the method of inductively coupled plasma atomic absorption spectrometry (ICPAES). The Hg concentration was determined with an automatic mercury analyzer RA-3 (NIC, Japan). In an attempt to ensure measurement accuracy and quality control, standard reference materials (GBW07401, Beijing, China) were consulted and the results showed that the analytical precision was within ±5% [27].

### 2.4. Contamination Evaluation

This study concerned itself with the evaluation of the accumulation of heavy metals in the riparian soils following an extended period of flooding. The metal concentrations in the soils were compared with their background values, referring to the samples taken from metal deposits in areas with a low level of human activity, to evaluate the pollution level of heavy metals. The regional heavy metals background concentrations in soils were obtained from the results of a national soil environmental background values investigation conducted in China by Wei [51], collecting a large number of samples and launching a statistical analysis of these samples (Table 1).

Considering the measured concentrations and background values, the geoaccumulation index (*I*_geo_), as proposed by Muller (1969), was used to evaluate the level of heavy metal contamination in the riparian soils by applying the following equation:(1)$$I_{\mathrm{geo}}=log_{2}\left(\frac{C_{i}}{1.5B_{i}}\right)$$

The examined concentration in soils of the metal element *i* is represented by *C_i_*, while *B_i_* is the corresponding environmental background concentration for element *i*. According to the values of the calculated *I*_geo_, the contamination level can be divided into seven degrees [52] (Table 2).

Furthermore, to evaluate the possible eco-toxicological effects of heavy metals in the soil on exposed organisms along the riparian zone, the potential ecological risk index (*RI*) [53] was also applied to assess the degree of heavy metal pollution. The *RI* is calculated by:(2)$$RI=\overset{n}{\underset{i=1}{\sum }}E_{ri}=\overset{n}{\underset{i=1}{\sum }}T_{ri}C_{f}^{i}=\overset{n}{\underset{i=1}{\sum }}T_{ri}\frac{C_{i}}{B_{i}}$$

The potential ecological risk coefficient is represented by *E_ri_* for the given metal *i*; *T_ri_* is the toxic-response factor for the given metal element *I*; *C_f_^i^* represents the contamination coefficient for a given metal *I*; and *C_i_* and *B_i_* represent the same meanings as Equation (1). *T_ri_* reflects the potential toxicity level for a given metal *i*. *T_ri_* is illustrated by Table 1.

### 2.5. Statistical Analysis

In order to compare the differences in the heavy metal content at different points, we counted the maximum and minimum value, mean value, and standard deviation of the heavy metal content at different sampling sections. Pearson’s correlation analysis was used to assess the relationships between different heavy metals.

## 3. Results

### 3.1. Distribution of Heavy Metal Concentrations in the Soil of the Riparian Zone

The concentrations of heavy metals in soils of the riparian zone displayed inconsistent modification trends with the decrease in elevation, whereas increasing trends were also observed (e.g., trace metals contents at the low attitudes for YBB, XWD, and ZJW) (Figure 3). In general, comparatively low heavy metals concentrations were detected in the soil of the riparian zone of XWR. In particular, the Cd concentration from all the sampling sites were lower than the regional background value. However, trace metals of As, Cr, Cu, Hg, Ni, Pb, and Zn at some sites were higher than the background value as shown in Appendix A. Specifically, the trace metals with higher concentrations than the local background contents were As from MJW and ZJW; Cr from YBB, MJW and HHR; Cu from YBB; Hg from ZJW; Ni from YBB and HHR; Pb from MJW; and Zn from MSB and HHR. Table 3 indicates that the transect with peak mean concentrations for As, Cd, Cr, Cu, Hg, Ni, Pb and Zn were ZJW, MSB, HHR, YBB, ZJW, HHR, MJW, and HHR, respectively. Furthermore, Appendix A demonstrates that the concentrations of As, Cr, Cu, Ni, Pb, and Zn at the XWD were lower than that of all the other sampling sites. Also, Figure 3 revealed that no significant difference was detected for the heavy metal concentrations of soils between the WLFZ and the IRZ. However, significant differences in the trace metals concertations along different sampling sections were observed (*p* < 0.01, Appendix A). Additionally, a significant difference in trace metal enrichment between LCR and HHR was detected (*p* < 0.01, Appendix A). Pearson’ s correlation analysis revealed many significant positive correlations for all tested metals in the soil, especially among As, Cu, and Zn with other metals (Table 3). The significant correlations for some trace metals indicates that elements are highly homologous.

### 3.2. Trace Metal Enrichment of Riparian Soils and Metal Contamination Assessment

Contamination status was determined by calculating the *I*_geo_ values of the riparian soils using Equation (1), with different pollution degrees. The results show that the pollution degree of measured heavy metals in the riparian soils of WXR were relatively low (Figure 4). *I*_geo_ revealed that soils in the riparian zone of XWR were not polluted by Cd, Cu, Ni, Zn, or Pb and was consistent for all the sampling sites. However, slight contamination by Hg at the transect of ZJW and Cr at the transect of HHR, and moderate contamination by As at the section of ZJW were monitored (Figure 4).

The contaminated trace metals introduce potential toxic risk to the local ecosystem. However, the risk changes considerably when the varying levels of toxicity in different metals are taken into account even though the exposure levels are similar [11]. Thus, the present study evaluated the potential toxicological effects of soil heavy metals on the riparian zone ecosystem using *RI* defined by Equation (2). The *E_r_* value for the individual metal element and the total *RI* value for all measured metals were calculated to obtain the status of potential ecological risk. The results showed that the *E_r_* values of Cd, Cr, Cu, Ni, Zn, and Pb in riparian soils are below 40 for all sampling transects and indicated a potentially minor risk; As and Hg followed a low-risk level at the transects of YBB, MSB, MJW, XWD, and HHR, whereas As is a moderate risk and Hg is a high risk at the ZJW (Figure 5). Overall, the soils in the riparian zone of YBB, MSB, MJW, XWD, and HHR represented a minor risk to the local ecosystems, whereas the riparian soils at ZJW denoted a moderate risk (Figure 6). Furthermore, Figure 6 also demonstrates that *RI* is gradually decreasing from the upstream to XWD, which denotes the corresponding ecological risk changing for riparian soils of XWR.

## 4. Discussion

### 4.1. Lateral and Longitudinal Distribution of Heavy Metal Concentrations in the Riparian Soils

Usually, the geochemical loads, including the dissolved and particulate forms in the river system can be physically retained in the reservoir. This can be accomplished by water impounding and sediment trapping from upstream and the reservoir catchment area due to the dam operation and reservoir regulation [3,11,54]. The present study determined the level of heavy metal concentrations in the soil of the riparian zone of XWR and evaluated its enrichment distributions at different sites. The investigation results revealed that most of the heavy metals present in the riparian soils did not fall within the denoting pollution status, although several metals fell within an evidently higher status compared to local background values of trace metals. These results suggest the contribution of an external source to the concentration of heavy metals during the alternation between wet and dry seasons of the XWR project. Since XWR is located in a mountainous valley area and artificial disturbances in the riparian zone are not strong, the increased heavy metal contained in soil can be attributed to the chemical transformation of soluble fractions in the reservoir water column during the alternation between wetting and drying [11]. The physical adsorption from the contaminated sediments may lead to an increase in trace metal concentrations in the soil of the riparian zone.

The concentrations of heavy metals in the riparian soils at different elevations, illustrated by Figure 3, showed a small lateral variation as evident from the trace metal concentrations, although no statistically significant trend is visible. This result is in accordance with the investigation conducted in the riparian zone of the Manwan Reservoir, Lancang River by Liu et al. [13] and the examination of the riparian zone located at the Three Gorges Reservoir by Wang et al. [55]. However, the results differ from the study by Tang et al. [54], which demonstrated that the heavy metal concentrations in the riparian soils of TGR generally follow a slight decreasing trend with an increase in elevation. Ye et al. [12] reported that the riparian soil of TGR was moderately contaminated by As and Pb according to several indices, and dam operation and local human activity affect heavy metal distribution in the riparian zone. The latest case study from Nuozhadu Reservoir, reported that the concentrations of Cu and Ni in the WLFZ soils increased noticeably with the increasing elevations, while Cr and Zn tended to decrease with increasing elevations [56]. The Nuozadu Reservoir is located at downstream of the XWR, approximately 300 km from XWD. The different accumulation and redistribution of trace metals in the riparian soils between the XWR and the Nuozadu Reservoir can partly be due to the different land-use types and inundations durations. Several soil properties differed significantly with land-use types in WLFZ of the Nuozhadu reservoir [56]. Whereas the land-use types in the WLFZ of XWR are simple, the sampling sites of XWD, MSB, ZJW, and HHR were forest before inundation and the sampling sites of YBB and MJW were farmland before inundation. The single type of land-use results in homogeneous soil properties in the riparian zone of XWR.

In summary, this study believes that the metal content of the soils in the riparian zone is less affected by the upstream incoming water and sediment, and mainly determined by the nature of the local soil, and is closely related to the background value. At the same time, the metal content of the soil in the fluctuation zone is related to the flooding intensity at different elevations to a certain extent.

### 4.2. Influence of the Changed Hydrologic Processes on Heavy Metals Accumulation in Riparian Soils

Dam operation and local anthropological activities affect the heavy metals distribution in the riparian zone [12]. Changes in the flooding duration owing to reservoir operation leads to heavy metal accumulation in both the sediments and soil of the riparian zone [11,57]. Inundation, as a result of water level regulations and natural flooding, has promoted heavy metal accumulation in the riparian soils and sediments of TGR due to the reservoir operation [9,11,27,54,58]. The uncommon result garnered from this study might either be attributed to the inadequate submerging time of the trace metals for any influence on the variations in the water levels to become apparent [55,59], or to the sampling elevation distance in this investigation being too small to reflect substantial distinctions [13]. The regulated operation of water levels in the reservoir also influenced the trace metal concentrations of the riparian soils and sediments. The water level of XWR was maintained at a relatively low elevation of 1180 m for flood prevention from April to May before the rainy season commenced. The relatively prolonged flooding residence time in the lower regions of the riparian zone contributed to further heavy metal accumulation, being noticeable in the riparian soils in this low-lying area. However, during the flood season, the contaminated sediments and water with high trace metal concentrations produced from towns and agricultural areas were enriched at the lower positions of WLFZ, especially at the beginning of the rainy season [11,58,60,61]. Thus, the diffuse chemical loads accumulated in the riparian soil can become more obvious during several heavy rainfalls. In contrast, the pure sediment from bank erosion and clear water from areas upstream during the dry season leads to the inconspicuous accumulation of trace metals in the riparian soil [11]. 

Furthermore, the significant differences in the trace metal concentrations observed among the different sampling belts (Appendix A), indicated the combined influence of human activities and natural factors on metal concentrations [11,18]. Considering the immovable attribute of soils in the riparian zone, trace metals accumulation in the soils can be explained by the chemisorption taking place from soluble portions in sediments and water from upstream. Additionally, vertical transfer of heavy metals from upper_layer sediments is partially responsible for heavy metal accumulation in the riparian soil [54]. However, the substantial difference in trace metal concentrations in soils of the riparian zone located at the tail, center, and head of XWR (Appendix A) may be related to the fluctuation intensity of the water levels and flow velocity in different sections of the reservoir. The end of the reservoir is close to the natural river channel (Figure 7) with a rapid water flow velocity, and it is difficult for the heavy metals carried in the water body and sediments to be absorbed by riparian soils. Thus, the proportion of heavy metals present in riparian soils due to fluvial processes such as river transport and sedimentation is relatively small in the reservoir tail. Conversely, the deceleration of flow velocity in the reservoir center along with a significant amount of sedimentation increases the contact opportunity and adsorption time for trace metals in the riparian soils. Therefore, the trace metal concentrations in soils of the riparian zone displayed an increasing tendency (Appendix A). Although the water flow rate slowed even further at the head of the reservoir, most of the sediment had already settled in the center of the reservoir. The heavy metals in the water are being diluted in the long-term hydrological process and, consequently, the riparian soil in the head area of the reservoir finds it challenging to adsorb trace metal elements from the sediments or inflows from upstream. Therefore, apparent heavy metals enrichment in the riparian soils at the head of XWR was not observed. Conversely, long-term water immersion leads to the release of heavy metals within dissolved portions in the soil and finally resulted in a decrease in the heavy metal contents. It is worth noting that the upper reaches of XWR, i.e., the Gongguoqiao Reservoir area, were severely polluted by the heavy metals Pb, Zn, Cd and As present in the soil, sediments, and water body. This contamination can be ascribed to the exploitation of the largest lead-zinc mining deposit in China, the Lanping Pb-Zn mines area in the Bijiang River basin [62]. Typically, the high-risk heavy metals contamination in the soils, sediments, and water of the river region upstream would lead to corresponding pollution in the downstream environmental mediums to varying degrees. However, the contamination assessments of *I*_geo_ and *RI* indicated that the riparian soils in the mainstream of LCR, from the reservoir tail to head, were not contaminated by the trace metals selected in this study (Figure 4&6). An important reason for this result is due to the Gongguoqiao Dam operation, which caused the chemical loads in both particulate and soluble forms. The chemical loads were physically retained during the hydrological regime by impounding water and trapping sediments from upstream and the catchment area of XWR.

A significant difference in trace metal enrichment was detected in LCR and HHR (Table 3). The concentrations of As and Hg in ZJW are significantly higher than the regional background value. Specifically, the rate of the measured value to the background was 4.56 for As and 2.59 for Hg, respectively. The trace metals of Cr, Ni and Zn also had relatively higher values than the local background values in HHR. These results implied the external input of contaminated water and sediments produced from the upper riparian zone or upstream artificial sources, for example, metal mining and smelting plants along the Heihui River, industrial wastewater (Heihui River is known as the “industrial corridor”), domestic sewage discharge and waste treatment from rural and urban settlements [13,40]. Furthermore, diffuse sources, including surface runoffs, dust diffusion of the road, atmospheric fallout, agricultural pollutants from fertilizer, pesticides and herbicides contributed to the contamination of soil in the riparian zone [11,13,40]. The development of fish farming in HHR after the XWR operation contributed to the relatively higher concentrations of As and Hg in ZJW and Pb, Ni, and Zn in HHR to a certain extent.

In conclusion, research into the discernible changes relating to the concentrations of heavy metals in the riparian soil of XWR was necessary to investigate the role of WLFs in combination with additional anthropogenic and natural factors. The results of the present study are beneficial in understanding the redistribution of heavy metals in the soil of the riparian zone of XWR as ascribed to the dam operation and reservoir filling. Heavy metal enrichment in the soils of the riparian zone is strongly associated with the water flow and sediments discharge from upstream. Hence, more detailed investigations are necessary to clarify the impact of WLFs on fluvial processes, especially the procedures of sedimentation, bank erosion, and chemical exchange of water_sediment_soils. The present results suggest that the dam closure and reservoir operation play a significant role in trace metals interception, although more specific evidence is needed to verify this.

## 5. Conclusions

Heavy metal contents were generally relatively low in the riparian soils of XWR and followed an inconsistent lateral decreasing trend, even though enriched concentrations were found for several metals. Significant differences in the trace metals concertations along different sampling sections were observed. Metal contamination assessment with *I*_geo_ indicated that the pollution degree of the measured heavy metals in the riparian soils of XWR were relatively low, while there were slightly polluted by Hg at the transect of ZJW and Cr at the transect of HHR, and moderately contaminated by As at the transect of ZJW. Cd, Cr, Cu, Ni, Zn, and Pb in riparian soils indicated a potentially minor risk for all sampling transects; As and Hg followed a low-risk level at the transects of YBB, MSB, MJW, XWD, and HHR, whereas As is a moderate risk and Hg is a high risk at the ZJW. The potential ecological risk index (*RI*) of the riparian soils revealed that As poses a moderate risk and Hg poses a high risk at ZJW. The XWR operating at full capacity has facilitated changes in the regional hydrological processes and the corresponding trace metals accumulation in the riparian zone. This study investigated the spatial pattern and contamination levels of soils in five typical sections of the riparian zone and confirmed their links to the altered water level processes. Polluted sediments and water introduced from upstream and upland contaminated sources are responsible for heavy metals accumulation in the riparian soils through chemisorption from solubilized fractions during the water_sediment_soils exchange processes following alternating wetting and drying due to reservoir operation. This study could assist in comprehending trace metal enrichment within the riparian soil of XWR attributed to the dam closure and reservoir impoundment. However, more rigorous examination is needed to elucidate the impact of WLFs on the chemical exchange of water_sediment_soils in the riparian zone.

## Figures and Tables

**Figure 1 ijerph-19-12902-f001:**
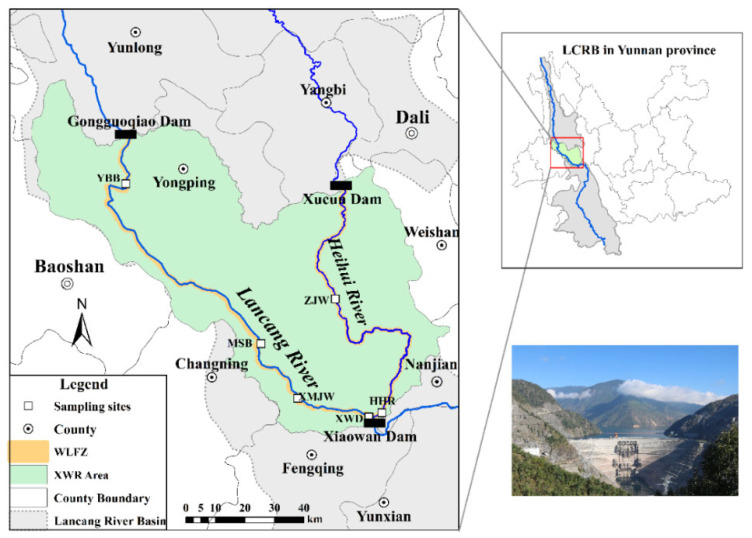
The Xiaowan Reservoir area in the Lancang River and the sampling locations.

**Figure 2 ijerph-19-12902-f002:**
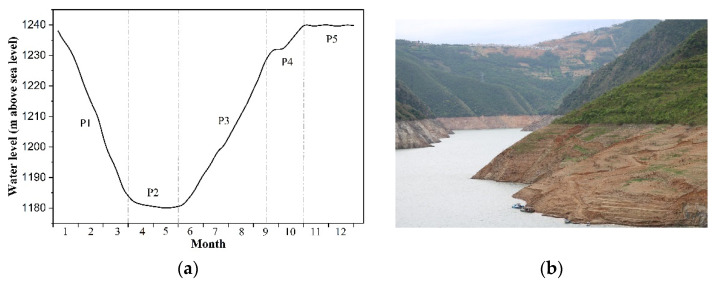
The changes in the water level by regular dam operation in the Xiaowan Reservoir. (**a**) Annual water level changes. (**b**) Water-level fluctuations zone.

**Figure 3 ijerph-19-12902-f003:**
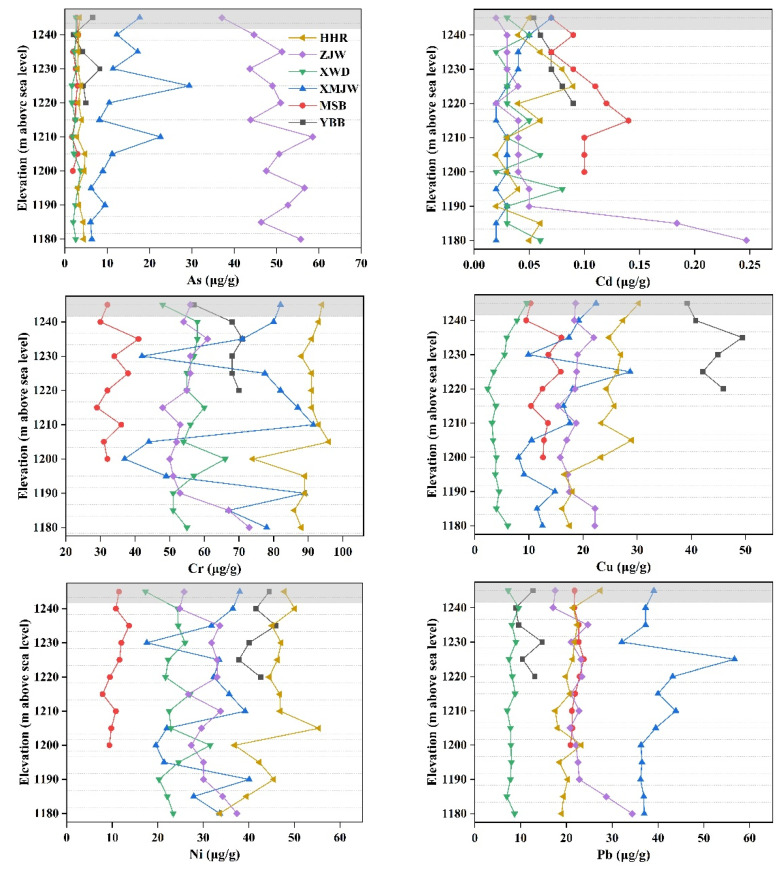
The lateral distribution of the heavy metal concentrations in the riparian soils at different sites: YBB, MSB, MJW, XWD, ZJW, and HHR. The plots with gray marks represent the infralittoral reference zone (IRZ).

**Figure 4 ijerph-19-12902-f004:**
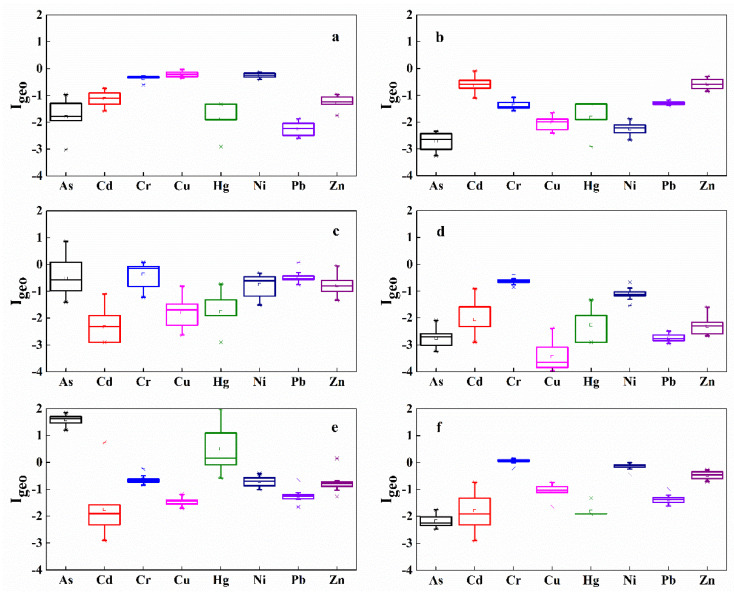
Box-whisker plots of *I*_geo_ values for different heavy metals in the riparian soils of XWR (The small letters (**a**–**f**) represent the same meaning as mentioned in Figure 3).

**Figure 5 ijerph-19-12902-f005:**
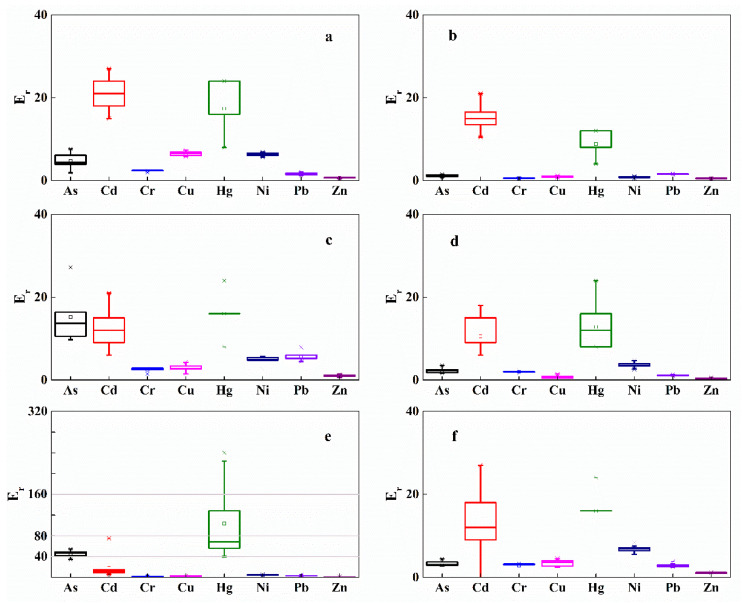
Box-whisker plots of *E_r_* values for different heavy metals in soils of the riparian zone (The small letters (**a**–**f**) represent the same meaning as mentioned in Figure 3).

**Figure 6 ijerph-19-12902-f006:**
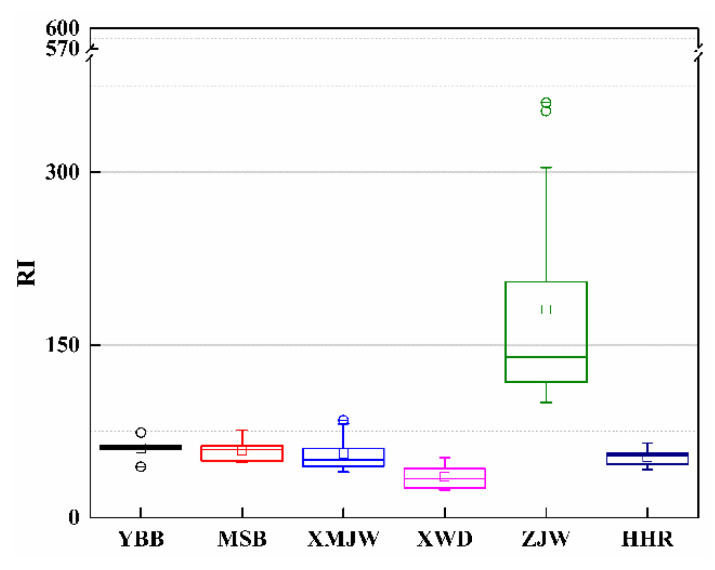
Box-whisker plots of *RI* values and the corresponding ecological risk for riparian soils at different sampling sites.

**Figure 7 ijerph-19-12902-f007:**
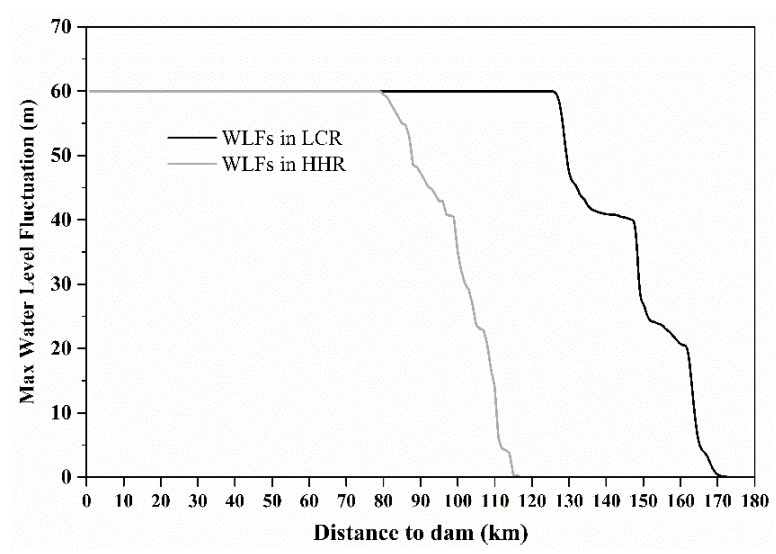
The maximum water level fluctuations in the Lancang River and Heihui River of XWR.

**Table 1 ijerph-19-12902-t001:** Geochemical background value and toxicity coefficient of heavy metals in soils of Yunnan Province, China.

Element	As	Cd	Cr	Cu	Hg	Ni	Pb	Zn
Background value (mg/kg)	10.8	0.1	57.6	33.6	0.05	33.4	36.0	80.5
Toxicity coefficient (*T_ri_*)	10	30	2	5	40	5	5	1

**Table 2 ijerph-19-12902-t002:** The calculated indexes responding to the contamination status and potential ecological risk levels.

*I*_geo_ Value	*I*_geo_ Class	Contamination Status	*E_ri_* Value	*RI* Value	Ecological Risk Level
≤0	0	UP	≤40	≤150	Low
0–1	1	UP-MP	40–80	150–300	Moderate
1–2	2	MP	80–160	300–600	High
2–3	3	MP-SP	160–320	>600	Very high
3–4	4	SP	>320		Extremely High
4–5	5	SP-EP			
>5	6	EP			

UP: Unpolluted; UP-MP: Unpolluted to moderately polluted; MP: Moderately polluted; MP-SP: Moderately to strongly polluted; SP: Strongly polluted; SP-EP: Strongly to extremely polluted; EP: Extremely polluted.

**Table 3 ijerph-19-12902-t003:** The correlation matrix amongst the heavy metal concentrations in the riparian soils.

	As	Cd	Cr	Cu	Hg	Ni	Pb	Zn
**As**	1							
**Cd**	−0.22	1						
**Cr**	0.18	−0.24 *	1					
**Cu**	0.43 **	0.21	0.56 **	1				
**Hg**	0.66 **	0.02	−0.05	0.27 *	1			
**Ni**	0.33 **	−0.095	0.91 **	0.77 **	0.14	1		
**Pb**	0.58 **	-.110	0.16	0.232*	0.24 *	0.11	1	
**Zn**	0.24*	0.36 **	0.33 **	0.60 **	0.20	0.36 **	0.57 **	1

* means the 0.05 level of significance, ** means the 0.01 level of significance.

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
