# Peer review of "Heavy Metals Enrichment Associated with Water-Level Fluctuations in the Riparian Soils of the Xiaowan Reservoir, Lancang River"

_ijerph, 2022, doi:10.3390/ijerph191912902_

Round 1
Reviewer 1 Report
Hydropower developments bring essential influences on the physical processes of the river banks, and thus may also impose effects on the pollution patterns. As a representative area, the study in the Xiaowan Reservoir makes great significances. The authors identified the typical heavy metals accumulation in the soils of the studied reparian zone which had been influenced by the altered hydrological regime, based on well-organized soil sampling, analysis and data interpretation. The manuscript was also well written. Overall, I think this article meets the interests of our readers and is recommended to be accepted for publication after minor revision.
Some suggestions for revision:
(1) Line 36: Please re-number the references according to the order of the first cite in the article.
(2) Line 49: "(" should be deleted.
(3) Fig. 2: Please add the caption of Fig. 2(b).
(4) Line 185: The chemicals should be correctly spelled. Please note the numbers.
(5) Table 1 and Table 2: The tables should be numbered according to the order that first cited in the article.
(6) A simple description of the statistical analysis for data processing should be added in the methods.
Author Response
Responses to comments from reviewer 1
Hydropower developments bring essential influences on the physical processes of the river banks, and thus may also impose effects on the pollution patterns. As a representative area, the study in the Xiaowan Reservoir makes great significances. The authors identified the typical heavy metals accumulation in the soils of the studied reparian zone which had been influenced by the altered hydrological regime, based on well-organized soil sampling, analysis and data interpretation. The manuscript was also well written. Overall, I think this article meets the interests of our readers and is recommended to be accepted for publication after minor revision.
Response: We appreciate your positive comments, thanks very much.
Some suggestions for revision:
(1) Line 36: Please re-number the references according to the order of the first cite in the article.
Response: We have re-number all the references according to the order of the first cite in the manuscript.
(2) Line 49: "(" should be deleted.
Response: We deleted this bracket.
(3) Fig. 2: Please add the caption of Fig. 2(b).
Response: We added the caption of Fig. 2(b), please see Figure 2 with revised version.
(4) Line 185: The chemicals should be correctly spelled. Please note the numbers.
Response: We have corrected this mistake. Please see line 185 in revised versio.
(5) Table 1 and Table 2: The tables should be numbered according to the order that first cited in the article.
Response: We have adjusted the position order of Table 1 and Table 2 according to the order of the first cite in the manuscript.
(6) A simple description of the statistical analysis for data processing should be added in the methods.
Response: We agree and added the 2.5 Statistical analysis in 2 Materials and Methods.

Reviewer 2 Report
The authors of the article entitled “Heavy metals enrichment associated with water-level fluctuations in the riparian soils of the Xiaowan Reservoir, Lancang River" presented research on, concentrations of the trace metals As, Cd, Cr, Cu, Hg, Ni, Pb and Zn in the riparian soils with assessing the geoaccumulation index (Igeo) and the ecological risk index (RI). Also, Authors assessed the relationship between heavy metals and water level fluctuations caused by the dam functioning.
After a thorough reading of the manuscript, there were mistakes that should be corrected in order for the article to be accepted for publication in the Journal.
1. The introduction should be expanded to include some of recent research issues and relevant literature, including metal contamination and migration of contaminants in the soil, together with possible threats to the environment.
2. Authors should pay attention to correctly cite the literature. In the text there should not be added the publication year of the article in parentheses but instead the reference number for example lines 308, 309, 310, 312. It applies to the entire manuscript.
3. In the section “Soil sampling and chemical analysis”, the authors should provide the purity of the acids used to obtain the mixture for samples mineralization, along with the names of manufacturers. In addition to the stated amounts of acids used to prepare the mixture, the ratio should also be given. The name of one of the three used acids cannot be expanded, only either all or none. The Authors should also include the ICP model and the name of the manufacturer used for the analyzes. The Authors need to explain for the readers what is the SEPAC method in line 187.
4. Throughout this article, Authors should change the measurement units from cm to m, an example is line 279.
5. Throughout the article, the formulas and abbreviations should be standardized so that they look the same, e.g. lines 18, 213 and 207 for Igeo, the same for, RI.
6. The whole text should be checked for errors regarding e.g. the lack of an author as in line 200.
7. The authors should change the graphical presentation of metal concentrations in such a way that they can be compared, in Figure 3, unfortunately, it is not possible to compare the elements with each other, but they were combined with each other at different scales, which is unacceptable.
8. For all drawings in the article, the sizes should be corrected and justified in relation to the text in accordance with the guidelines in the MS template from the IJERPH. When dividing the figure into several parts, each a, b, c ... should be explained in the description of the figure. For figures 5 and 6, the abbreviations used should be explained in the description below the figure.
9. In Figure 2 and in the text, authors should indicate the unit in e.g. “m above sea level”.
10. The whole article should also be corrected in terms of repeating words, e.g. “generally”.
11. In Table 3, the font for the Palatino Linotype used by MDPI should be unified.
12. Figure 4 is difficult to read and should be presented in a different form.
13. The conclusions require improvement and the provision of specific information on the basis of the research and analysis carried out and translating these results into practice.
14. References should also be corrected. If there is more than one author of the publication, all authors should be listed in accordance with the MDPI requirements, which are also presented in the MS template in the IJERPH.
15. Punctuation and editorial corrections as well as minor linguistic corrections are necessary throughout the article. Any unnecessary spaces should be corrected (applies to the entire article), an example is line 185.
After the above-mentioned comments are corrected, the article will meet the formal requirements to be published in the International Journal of Environmental Research and Public Health, as it is also included in the aims and scope of the Journal.
Suggested literature:
Wang, L.K.; Veysel, E.; Ferruh, E. Handbook of Advanced Industrial and Hazardous Wastes Treatment; CRC Press, Taylor & Francis Group: Boca Raton, FL, USA, 2009.
Tack, F.M.G.; Bardos, P. Overview of Soil and Groundwater Remediation. In Soil and Groundwater Remediation Technologies; Ok, Y.S., Rinklebe, J., Hou, D., Tsang, D.C.W., Tack, F.M.G., Eds.; Taylor & Francis: Oxfordshire, UK, 2020.
Pusz, A.; Wiśniewska, M.; Rogalski, D. Assessment of the Accumulation Ability of Festuca rubra L. and Alyssum saxatile L. Tested on Soils Contaminated with Zn, Cd, Ni, Pb, Cr, and Cu. Resources, 2021, 10 (5), 1–18.
Saha, J.K.; Selladurai, R.; Coumar, M.V.; Dotaniya, M.L.; Kundu, S.; Patra, A.K. Assessment of Heavy Metals Contamination in Soil. In Soil Pollution—An Emerging Threat to Agriculture; Environmental Chemistry for a Sustainable World; Springer: Singapore, 2017; Volume 10.
Author Response
Responses to comments from reviewer 2
The authors of the article entitled “Heavy metals enrichment associated with water-level fluctuations in the riparian soils of the Xiaowan Reservoir, Lancang River" presented research on, concentrations of the trace metals As, Cd, Cr, Cu, Hg, Ni, Pb and Zn in the riparian soils with assessing the geoaccumulation index (Igeo) and the ecological risk index (RI). Also, Authors assessed the relationship between heavy metals and water level fluctuations caused by the dam functioning.
After a thorough reading of the manuscript, there were mistakes that should be corrected in order for the article to be accepted for publication in the Journal.
Response: Thanks for your comments. We addressed the concerns and revised the manuscript based on your insightful suggestions and comments.
- The introduction should be expanded to include some of recent research issues and relevant literature, including metal contamination and migration of contaminants in the soil, together with possible threats to the environment.
Response: Thanks for your suggestion, however, the introduction is liittle long, and the topic of this study was the water-level fluctuations effect the heavy metals accumulation in soils of the riparian zone. So we revised the introdution but not expand.
- Authors should pay attention to correctly cite the literature. In the text there should not be added the publication year of the article in parentheses but instead the reference number for example lines 308, 309, 310, 312. It applies to the entire manuscript.
Response: We agree and correct all the cite literatures in the text and references.
- In the section “Soil sampling and chemical analysis”, the authors should provide the purity of the acids used to obtain the mixture for samples mineralization, along with the names of manufacturers. In addition to the stated amounts of acids used to prepare the mixture, the ratio should also be given. The name of one of the three used acids cannot be expanded, only either all or none. The Authors should also include the ICP model and the name of the manufacturer used for the analyzes. The Authors need to explain for the readers what is the SEPAC method in line 187.
Response: Thanks for your thorough comment. We added the purity of the acids used; however, we can’t provide the names of manufacturers for the used acids and the ICP due to these lab test has been conducted in 2018. And we added the the ratio for different acids. The SEPAC method in line 187 is the experimental method recommended by the State Environmental Protection Administration of China.
- Throughout this article, Authors should change the measurement units from cm to m, an example is line 279.
Response: We agree and checked all measurement units in the text. However, the comment mentioned “an example is line 279”, we guess your mean in line 179, here “0 to 10 cm” refer to the soil samping depth.
- Throughout the article, the formulas and abbreviations should be standardized so that they look the same, e.g. lines 18, 213 and 207 for Igeo, the same for, RI.
Response: We agree and standardized all the formulas and abbreviations throughout the article. Please see the revised manuscript.
- The whole text should be checked for errors regarding e.g. the lack of an author as in line 200.
Response: We agree and checked all the errors in the text, e.g., added an author as in line 200.
- The authors should change the graphical presentation of metal concentrations in such a way that they can be compared, in Figure 3, unfortunately, it is not possible to compare the elements with each other, but they were combined with each other at different scales, which is unacceptable.
Response: We agree and modified the Figure 3 with line +scatter for better understand the metal concentrations in such a way that they can be compared. Please check in the revised manuscript.
- For all drawings in the article, the sizes should be corrected and justified in relation to the text in accordance with the guidelines in the MS template from the IJERPH. When dividing the figure into several parts, each a, b, c ... should be explained in the description of the figure. For figures 5 and 6, the abbreviations used should be explained in the description below the figure.
Response: We agree and corrected and justified the sizes in accordance with the guidelines in the MS template from the IJERPH. And we added the description for dividing figure, e. g., Figure 2. Also, we explained the abbreviations used in figures 5 and 6 below the figure.
- In Figure 2 and in the text, authors should indicate the unit in e.g. “m above sea level”.
Response: We agree and added the unit “m above sea level” In Figure 2 and in the text.
- The whole article should also be corrected in terms of repeating words, e.g. “generally”.
Response: Thanks for your suggestion, we agree and instead of repeating words, e.g., instead of “Generally” with “Theoretically” in line 68, instead of “Generally” with “Overall” in line 286
- In Table 3, the font for the Palatino Linotype used by MDPI should be unified.
Response: We agree and unified the the font for Table with the Palatino Linotype used by MDPI.
- Figure 4 is difficult to read and should be presented in a different form.
Response: Figure 4 presented the Box-whisker plots of Igeo values for different heavy metals in riparian soils of XWR, and the small letters (a, b, c, d, e and f) represent the different sampling sites.
- The conclusions require improvement and the provision of specific information on the basis of the research and analysis carried out and translating these results into practice.
Response: We agree and added specific information on the basis of the research and analysis carried out and translating these results into practice. Please see the conclusion in revised manuscript for details.
- References should also be corrected. If there is more than one author of the publication, all authors should be listed in accordance with the MDPI requirements, which are also presented in the MS template in the IJERPH.
Response: We agree and correct all the cite literatures format in references in accordance with the MDPI requirements.
- Punctuation and editorial corrections as well as minor linguistic corrections are necessary throughout the article. Any unnecessary spaces should be corrected (applies to the entire article), an example is line 185.
Response: Thanks for your suggestion, we agree and checked and corected some punctuation and editorial corrections as well as minor linguistic corrections throughout the article.
After the above-mentioned comments are corrected, the article will meet the formal requirements to be published in the International Journal of Environmental Research and Public Health, as it is also included in the aims and scope of the Journal.
Response: Thanks for your comments. We addressed the concerns and revised the manuscript based on your insightful suggestions and comments.
Suggested literature:
(1)Wang, L.K.; Veysel, E.; Ferruh, E. Handbook of Advanced Industrial and Hazardous Wastes Treatment; CRC Press, Taylor & Francis Group: Boca Raton, FL, USA, 2009.
(2)Tack, F.M.G.; Bardos, P. Overview of Soil and Groundwater Remediation. In Soil and Groundwater Remediation Technologies; Ok, Y.S., Rinklebe, J., Hou, D., Tsang, D.C.W., Tack, F.M.G., Eds.; Taylor & Francis: Oxfordshire, UK, 2020.
(3)Pusz, A.; Wiśniewska, M.; Rogalski, D. Assessment of the Accumulation Ability of Festuca rubra L. and Alyssum saxatile L. Tested on Soils Contaminated with Zn, Cd, Ni, Pb, Cr, and Cu. Resources, 2021, 10 (5), 1–18.
(4)Saha, J.K.; Selladurai, R.; Coumar, M.V.; Dotaniya, M.L.; Kundu, S.; Patra, A.K. Assessment of Heavy Metals Contamination in Soil. In Soil Pollution—An Emerging Threat to Agriculture; Environmental Chemistry for a Sustainable World; Springer: Singapore, 2017; Volume 10.
Response: Thanks for your suggested literature. We have discussed your suggestion carefully, and according to the topic of our paper, we think that reference 4 is more appropriate and added in the text and references.
